# Glucose-stimulated KIF5B-driven microtubule sliding organizes microtubule networks in mouse pancreatic β cells

Kai M Bracey[1], Margret A Fye[1], Alisa Cario[1], Kung-Hsien Ho[1,2], Pi'illani Noguchi[1], Guoqiang Gu[1,2,3], Irina Kaverina[1,3]*

[1]Department of Cell and Developmental Biology, Vanderbilt University, Nashville, United States; [2]Center for Stem Cell Biology, Vanderbilt University, Nashville, United States; [3]Program of Developmental Biology, Vanderbilt University, Nashville, United States

*For correspondence:
irina.kaverina@vanderbilt.edu

Competing interest: The authors declare that no competing interests exist.

## eLife Assessment

In their **valuable** study, Bracey et al. investigate how microtubule organization within pancreatic islet beta cells supports optimal insulin secretion. Using a combination of live imaging and photo-kinetic assays in an in vitro culture system, they provide **compelling** evidence that kinesin-1-mediated microtubule sliding, which plays critical roles in neurons and embryos, also plays a critical role in forming the sub-membranous microtubule band in response to glucose in beta cells. This work will be of interest to cell biologists studying cytoskeletal dynamics and organelle trafficking, as well as to translational biologists focused on diabetes.

**Abstract** In pancreatic islet β cells, molecular motors use cytoskeletal polymers microtubules as tracks for intracellular transport of insulin secretory granules. The β-cell microtubule network has a complex architecture and is nondirectional, which provides insulin granules at the cell periphery for rapid secretion response, yet to avoid oversecretion and subsequent hypoglycemia. We have previously characterized a peripheral sub-membrane microtubule array, which is critical for the withdrawal of excessive insulin granules from the secretion sites. Microtubules in β cells originate at the Golgi in the cell interior, and how the peripheral array is formed remains unknown. Here, we demonstrate that kinesin KIF5B, a motor protein with the capacity to transport microtubules as cargos, is needed to align sub-membrane microtubules in clonal mouse β cells MIN6 and β cells within intact mouse islets. Real-time imaging and photokinetics approaches indicate that KIF5B actively slides existing microtubules to the β-cell periphery. Moreover, like many physiological β-cell features, microtubule sliding is facilitated by a high glucose stimulus. These new data, together with our previous report that high glucose destabilizes the sub-membrane microtubule array to allow for robust secretion, indicate that MT sliding is another integral part of glucose-triggered microtubule remodeling, likely replacing destabilized peripheral microtubules to prevent their loss over time and β-cell malfunction.

## Introduction

The precise level of glucose-stimulated insulin secretion (GSIS) from pancreatic β cells is crucial for glucose homeostasis. On one hand, insufficient insulin secretion decreases glucose uptake by peripheral tissues, leading to diabetes. On the other hand, excessive secretion causes glucose depletion from the bloodstream and hypoglycemia. Not surprisingly, multiple levels of cellular regulation control the amount of insulin secretory granules (IGs) released on every stimulus. One level of this control is

facilitated by microtubules (MTs), intracellular polymers that serve as tracks for intracellular transport of IGs and define how many IGs are positioned at the secretion sites (*Desai and Mitchison, 1997*; *Heaslip et al., 2014*; *Varadi et al., 2002*). Thus, the architecture and dynamics of the MT network are instrumental for secretion control.

The findings in the last decade have uncovered that the organization and regulation of the β-cell MT network are quite unusual. As in several other eukaryotic cells, MTs in β cells are nucleated at MT-organizing centers (MTOCs) in the cell interior, partially at the centrosome and to a large extent at the Golgi membranes (*Trogden et al., 2019*; *Zhu et al., 2015*). Conventionally, this should be followed by MT plus-end polymerization toward the cell periphery, resulting in a radial MT array with high MT density in the center rather than in the periphery. However, the β cell lacks such MT polarity, well characterized in mesenchymal cells in culture (*Bracey et al., 2022*). Instead, interior β-cell MTs are twisted and interlocked (*Varadi et al., 2003*; *Zhu et al., 2015*), while peripheral MTs are arranged in a prominent array of co-aligned and often bundled MTs underlying the cell membrane, hereafter called the sub-membrane MT array or the peripheral MT array (*Bracey et al., 2020*).

Like other features on β-cell physiology, MTs in β cells are regulated by glucose metabolism. As protein polymers, MTs are capable of regulated polymerization and depolymerization, which allows for fine-tuning of their organization. Under basal glucose conditions, β-cell MTs are stable and undergo little turnover (*Ho et al., 2020*; *Zhu et al., 2015*). Sub-membrane MT arrays are especially highly stabilized by an MT-associated protein (MAP) tau, well known for its neuronal functions (*Ho et al., 2020*). Upon a high glucose stimulus, tau is phosphorylated and sub-membrane MTs undergo destabilization (*Ho et al., 2020*) and fragmentation, possibly via an MT-severing activity (*Müller et al., 2021*). MT destabilization is accompanied by facilitated formation of new MTs at the main β-cell MTOC, the Golgi (*Trogden et al., 2019*), and increased MT polymerization (*Heaslip et al., 2014*), which are thought to replace destabilized MTs and restore the MT network.

Such complex, glucose-dependent organization of MT networks in β cells is thought to underly a multifaceted involvement of MT transport in insulin secretion regulation: MTs regulate the availability of IGs for secretion in both positive and negative fashion. First, Golgi-derived MTs are necessary for efficient IG generation at the trans Golgi network (*Trogden et al., 2019*). Second, interior MTs are to a large extent responsible for IG transportation throughout the cell, which occurs predominantly in the nondirectional, diffusion-like manner (*Tabei et al., 2013*; *Zhu et al., 2015*), likely due to the interlocked configuration of the interior MT network. In addition, directional MT-dependent runs of secretion-competent granules toward the cell periphery have been described (*Hoboth et al., 2015*; *Müller et al., 2021*). These processes contribute to the positive regulation of GSIS by preparing and distributing IG. At the same time, sub-membrane MT arrays serve for withdrawal of excessive peripheral IGs from the secretion sites, which prevents IG docking and acute oversecretion upon a given stimulus (*Bracey et al., 2020*; *Hu et al., 2021*; *Zhu et al., 2015*). The latter process provides a negative MT regulation of secretion.

During GSIS, the abovementioned glucose-dependent destabilization of the sub-membrane MT array must reduce IG withdrawal and downplay the negative MT-dependent regulation of secretion (*Bracey et al., 2020*). At the same time, new MTs polymerizing off the Golgi facilitate IG biogenesis and provide material to rebuild destabilized network (*Trogden et al., 2019*). This is likely preparing cells for the next round of stimulation. Thus, existing data provide at least initial understanding of the mechanisms whereby β-cell MT architecture allows for fine-tuning of GSIS levels.

However, it is yet unclear how the complex β-cell MT network forms. Being nucleated at MTOCs in the cell interior, it is puzzling that the resulting β-cell MTs are not organized in conventional radial arrays and rather form a prominent peripheral array (*Bracey et al., 2020*). The goal of this study is to uncover the mechanisms underlying the development of β-cell-specific MT network.

One of the established ways to modify the MT network without changing the location of MTOCs is to relocate already polymerized MTs by active motor-dependent transport. This phenomenon is called 'MT sliding' (*Straube et al., 2006*). Several MT-dependent molecular motors have been implicated in driving MT sliding (*Lu and Gelfand, 2017*). In some cases, a motor facilitates MT sliding by walking along a MT while its cargo-binding domain is stationary being attached to a relatively large structure, e.g., plasma membrane. This causes sliding of an MT which served as a track for the stationary motor. This mechanism has been described for dynein-dependent MT sliding (*Grabham et al., 2007*; *He et al., 2005*). MTs can also be efficiently slid by motors which have two functional motor assemblies,

such as a tetrameric kinesin-5/Eg5 (*Acar et al., 2013*; *Vukušić et al., 2021*), or which carry an MT as a cargo while walking along another MT. For the latter mechanism, a motor needs a non-motor domain with a capacity to bind either an MT itself or an MAP as an adapter (*Cao et al., 2020*; *Kurasawa et al., 2004*; *Vukušić et al., 2021*).

Out of these MT sliding factors, kinesin-1 is known to be critical for organizing unusual MT architecture in specialized cells. In oocytes, kinesin-1-dependent MT sliding empowers cytoplasmic streaming (*Lu et al., 2016*). In differentiating neurons, kinesin-1 moves organelles and MTs into emerging neurites, which is a defining step in developing branched MT networks and long-distance neuronal transport (*Jolly et al., 2010*; *Lu et al., 2013*). With these data in mind, kinesin-1 presents itself as the most attractive candidate for organizing MTs in β cells. This motor is highly expressed in β cells and is well known to act as a major driving force in IG transport and GSIS (*Meng et al., 1997*, *Varadi et al., 2002*, *Varadi et al., 2003*, *Cui et al., 2011*).

Here, we show that KIF5B, being the major variant of kinesin-1 in β cells, actively slides MTs in this cell type. We show that this phenomenon defines MT network morphology and supplies MTs for the sub-membrane array. Moreover, we find that MT sliding in β cells is a glucose-dependent process and thus likely participates in metabolically driven cell reorganization during each secretion cycle.

## Results

### Identification of KIF5B as the MT sliding motor in β cells

To address the factors that shape the configuration of MT networks in β cells, we tested for a potential involvement of motors-dependent MT sliding. Not surprisingly, analysis of existing RNA-sequencing data in functional mouse islet β cells highlighted kinesin-1 KIF5B as the highest expressing β-cell motor protein (*Figure 1A*; *Sanavia et al., 2021*). Since this kinesin has been reported to have MT sliding activity in many types of interphase cells, we tested its potential ability to slide MTs in β cells.

Efficient depletion of KIF5B was achieved by utilizing two independent lentiviral-based shRNA against mouse *Kif5b* in mouse insulinoma cell line MIN6 (*Figure 1B*, *Figure 1—figure supplement 1A*). To visualize MT sliding, shRNA-treated MIN6 cells expressing mEmerald-tubulin were imaged by live-cell spinning disk confocal microscopy. We photobleached MTs in two large cell regions, leaving a thin unbleached band ('fluorescent belt') and analyzed relocation of MTs from the 'fluorescent belt' into the bleached areas over time. To minimize the effects of plausible MT polymerization and to reduce photobleaching, MTs were imaged for short time periods (5 min). Strikingly, in control cells (treated with scrambled control shRNA), MTs were efficiently translocated from the 'fluorescent belt' into the photobleached area, indicating that MT sliding events are prominent in this cell type (*Figure 1C and D*). In contrast, MIN6 cells expressing either *Kif5b* shRNA variants displayed a significant loss of MT sliding ability (*Figure 1C, E, and F*, *Figure 1—video 1* 'MT sliding FRAP'), indicating that the loss of KIFB leads to the loss of MT sliding.

While the assay described above provides an easy visualization of MT sliding, it allows for visualization of only a subset of the MT network. To further corroborate the above findings, we used a less photodamaging system to visualize MT sliding that does not involve photobleaching and allows for evaluation of displacements within the whole MT network. To this end, we applied an MT probe of fiducial marks, K560Rigor$^{E236A}$-SunTag (*Tanenbaum et al., 2014*) in MIN6 cells (*Figure 1G–I*). This probe contains the human kinesin-1 motor domain (residues 1–560) with a rigor mutation in the motor domain (K560Rigor$^{E236A}$) and fused to 24 copies of a GCN4 peptide. The rigor mutation in the motor domain causes it to bind irreversibly to MTs (*Rice et al., 1999*). When co-expressed with a pHalo-tagged anti-GCN4 single-chain antibody (ScFv-GCN4-HaloTag-GB1-NLS), K560Rigor$^{E236A}$ can recruit up to 24 of the Halo ligands to a single position on an MT. The pHalo-tagged anti-GCN4 construct also contains a nuclear localization signal (NLS), which lends itself to reduce the background of the unbound dye. This enables visualization of MT sliding events via single molecule tracking of the fiducial marks along the MT lattice, allowing us to analyze MT sliding behavior within the whole network with high temporal and spatial resolution (*Figure 1G–K*, *Figure 1—figure supplement 1B*, *Figure 1—video 2* 'MT sliding Suntag').

Our data indicate that in cells treated with scrambled control shRNA, a subset of K560Rigor$^{E236A}$-SunTag fiducial marks underwent rapid directional movements, interpreted as MT sliding events (*Figure 1G and K*). In contrast, in cells expressing *Kif5b*-specific shRNAs, the fraction of rapidly moving

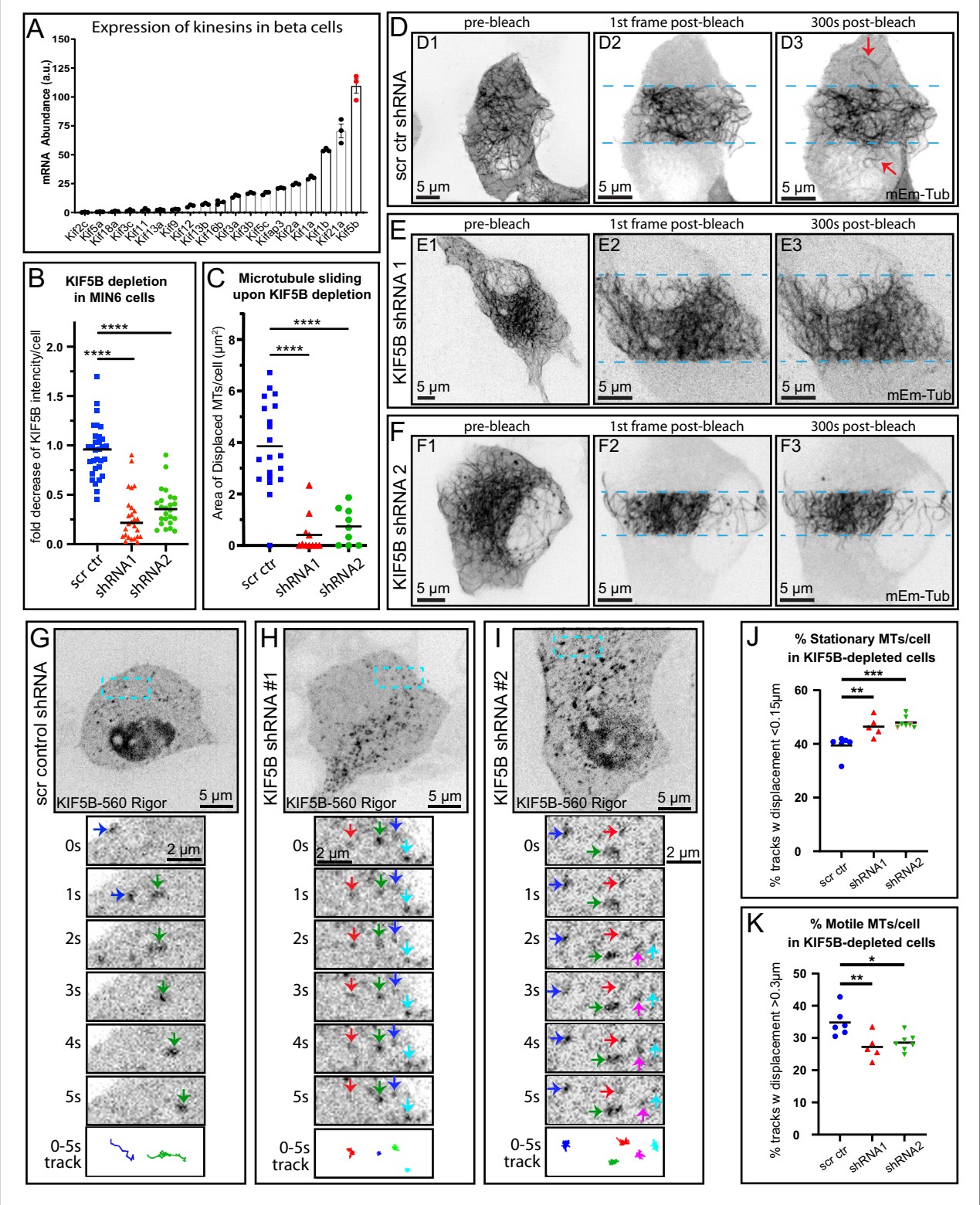

**Figure 1.** Microtubules (MTs) in pancreatic β cells undergo extensive sliding driven by kinesin KIF5B. (**A**) A subset of RNA-sequencing data from primary mouse β cells showing highly expressed kinesins as indicated by mRNA counts. KIF5B (most-right bar, red data points) is the most abundant kinesin motor in this cell type. N=3. Note that this is a subset of the RNA-sequencing sets published in *Sanavia et al., 2021*. (**B**) Efficient depletion of KIF5B in MIN6 cells using two alternative shRNA sequences, as compared to a scrambled shRNA control. Based on immunofluorescent staining of KIF5B as in *Figure 1—figure supplement 1* (**A–C**). Fold decrease of fluorescence signal per cell normalized to cells w/o shRNA expression in the same field of

*Figure 1 continued on next page*

*Figure 1 continued*

view. N=25–32 cells from 4 repeats. (C) Quantification of MT sliding FRAP assay in cells treated with scrambled control or one of the two KIF5B-specific shRNAs (see representative data in **D–F**). MT displacement is shown as the area of MTs displaced into the bleached area after 5 min of recovery. One-way ANOVA test was performed for statistical significance (p-value<0.0001). N=9–20 cells per set. (D–F) Frames from representative FRAP live-cell imaging sequences. mEmerald-tubulin-expressing MIN6 cells. Inverted grayscale images of maximum intensity projections of spinning disk confocal microscopy stacks over a 1-μm-thick ventral cell layer. Scale bars, 5 μm. (D1–F1) Overview of the whole cell prior to photobleaching. (D2–F3) Enlarged areas from (D1–F1) immediately after photobleaching (D2–F2) and 5 min (300 s) after photobleaching (D3–F3). Light-blue dotted lines indicate the edges of the photobleached areas. Red arrows indicate MTs displaced into the bleached area. Scale bars, 5 μm (**D–F**, *Figure 1—video 1* 'MT sliding FRAP'). (G–I) MIN6 cells featuring fiducial marks at MTs due to co-expression of SunTag-KIF5B-560Rigor construct and Halo-SunTag ligand. Representative examples for scrambled control shRNA-treated cell (G), KIF5B shRNA #1-treated cell (H), and KIF5B shRNA #2-treated cell (I) are shown. Single slices by spinning disk confocal microscopy. Halo-tag signal is shown as an inverted grayscale image. Top panels show cell overviews (scale bars, 5 μm). Below, boxed insets (scale bars, 2 μm) are enlarged to show dynamics of fiducial marks (color arrows) at 1 s intervals (1–5 s). 0–5 s tracks of fiducial mark movement are shown in the bottom panel, each track color-coded corresponding to the arrows in the image sequences. (J) Summarized quantification of stationary fraction of fiducial marks along MT lattice (5 s displacements below 0.15 μm). Scrambled shRNA control N=1421 tracks across 6 cells, shRNA#1 N=852 tracks across 5 cells, shRNA#2 N=2182 tracks across 7 cells. One-way ANOVA, p<0.001. (K) Summarized quantification of motile fraction of fiducial marks along the MT lattice (5 s displacements above 0.3 μm). Scrambled shRNA control N=2066 tracks across 6 cells, shRNA#1 N=390 tracks across 5 cells, shRNA#2 N=412 tracks across 7 cells. One-way ANOVA, p<0.001 (**G–I**, *Figure 1—video 2* 'MT sliding SunTag').

The online version of this article includes the following video, source data, and figure supplement(s) for figure 1:

**Source data 1.** SunTag marks displacement 5 s intervals across each cell.

**Figure supplement 1.** KIF5B depletion controls.

**Figure 1—video 1.** Microtubules (MT) sliding FRAP.

https://elifesciences.org/articles/89596/figures#fig1video1

**Figure 1—video 2.** Microtubule (MT) sliding SunTag.

https://elifesciences.org/articles/89596/figures#fig1video2

fiducial marks was significantly reduced, while the fraction of stationary fiducial marks increased (*Figure 1H, I, and J*), indicating the suppression of MT sliding. Collectively, these results indicate that KIF5B is necessary for MT sliding in MIN6 cells.

## KIF5B is required for β-cell MT organization

Because MT sliding mediated by KIF5B is a prominent phenomenon in β cells, we sought to test whether it has functional consequences for MT networks in these cells. Tubulin immunostaining revealed striking differences in MT organization between MIN6 cells treated with scrambled control shRNA versus *Kif5b*-specific shRNAs. While control cells had convoluted non-radial MTs with a prominent sub-membrane array, typical for β cells (*Figure 2A*), KIF5B-depleted cells featured extra-dense MTs in the cell center and sparse receding MTs at the periphery (*Figure 2B and C*). Significant reduction of tubulin staining intensity at the cell periphery (*Figure 2D*) confirms the robustness of this phenotype in MIN6 cells.

We proceeded to investigate MT organization within primary β cells of the *Kif5b* KO mouse line (*Kif5b^fl/−^:RIP2-Cre*, *Figure 2—figure supplement 1*). These mice exhibit a β-cell-specific deletion of KIF5B, as previously outlined (*Cui et al., 2011*). Our examination revealed a consistent presence of a significant peripheral array in the C57BL/6J control mice, while the KO counterparts exhibited a partial loss of this peripheral bundle. Specifically, the measured tubulin intensity at the cell periphery was significantly reduced in the KO mice compared to their wild-type (wt) counterparts (*Figure 2—figure supplement 1A–C*). Distinct extra-dense MTs were not common in islet β cells, possibly due to the 3D shape of a β cell in tissue and/or compensatory mechanisms in organisms. Thus, our results indicate that loss of KIF5B leads to a strong defect in MT location to the cell periphery in both a β-cell culture model and primary β cells within intact islets, which could be a direct consequence of disrupted sliding of MTs from the cell center to the periphery.

An alternative explanation for the loss of peripheral MTs would be the loss of the mechanical modification of the MT lattice by the force provided by moving kinesin motors. In cells with radial MT organization, this mechanism was shown to promote MT rescues and growth toward the cell periphery (*Andreu-Carbó et al., 2022*). To test the contribution of kinesin motor movement at MTs to MT density at the cell periphery, we have attempted to rescue the KIF5B-depletion phenotype by re-expressing kinesin-1 motor (see *Figure 3A-1*), which is capable of moving along MT but is unable

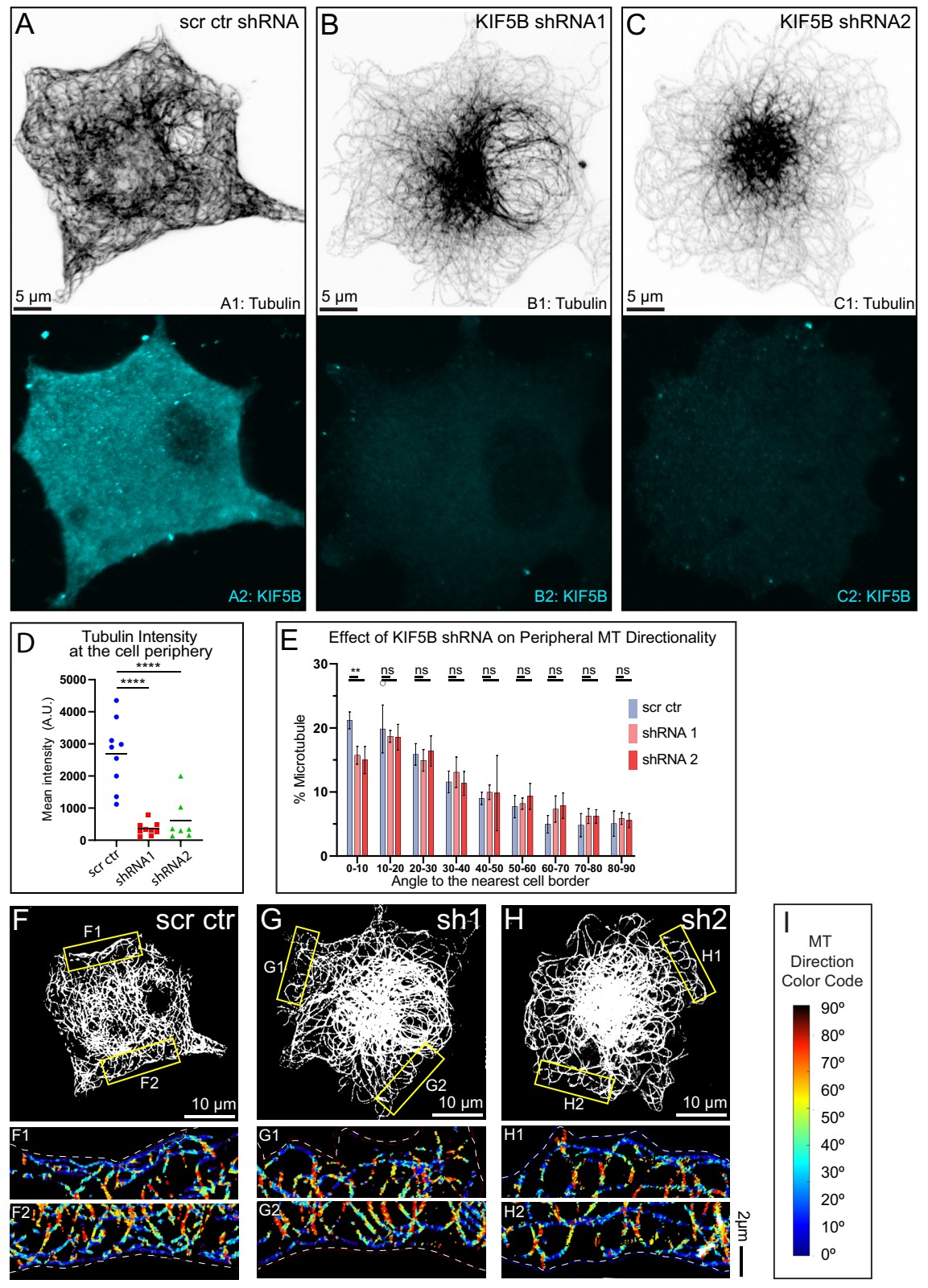

**Figure 2.** Microtubule (MT) abundance and alignment at the cell periphery depend on KIF5B. (**A–C**) MT organization in MIN6 cells expressing scrambled control shRNA (**A**), KIF5B-targeting shRNA #1 (**B**), or KIF5B-targeting shRNA #2 (**C**). Top, immunofluorescence staining for tubulin (grayscale, inverted). Bottom, immunofluorescence staining for KIF5B (cyan). Maximum intensity projection of 1 μm at the ventral side of the cell. N=12. Scale bars, 5 μm. (**D**) Quantification of mean tubulin intensity within the outer 2 μm peripheral area of a cell, in data represented in (**A–C**). Mean values, black bars.

*Figure 2 continued on next page*

*Figure 2 continued*

One-way ANOVA, p<0.0001. N=7–9 cells. (**E**) Histograms of MT directionality within 1 µm of cell boundary using perfected thresholds (see ***Figure 2— figure supplements 3 and 4*** for the analysis workflow and variants) in cells treated with scrambled control versus KIF5B-targeting shRNA. Data are shown for the summarized detectable tubulin-positive pixels in the analyzed single confocal slices of shRNA-treated cell population immunostained for tubulin, as represented in (**F–H**). Unpaired t-tests were performed across each bin for all cells, and a Kolmogorov-Smirnov (K-S) test was performed on the overall distribution. The share of MTs parallel to the edge (bin 0–10) is significantly higher in control as compared to KIF5B depletions. Pixel numbers in the analysis: SCR N=106,780 pixels across 9 cells, shRNA #1 N=71,243 across 7 cells, shRNA #2 N=60,087 across 7 cells. (**F–H**) Representative examples of MT directionality analysis in single confocal slices of shRNA-treated cells immunostained for tubulin, as quantified in (**E**). Single laser scanning confocal microscopy slices. (**F**) Scrambled control shRNA-treated cell. (**G**) KIF5B shRNA#1-treated cell. (**H**) KIF5B shRNA#1-treated cell. Overviews of cellular MT networks are shown as a threshold to detect individual peripheral MTs (see ***Figure 2—figure supplement 3***, panel A5). (F1– H2) Directionality analysis outputs of regions from yellow boxes in (**F–H**) are shown color-coded for the angles between MTs and the nearest cell border. (**I**) Color code for (F1–H2): MTs parallel to the cell edge, blue; MTs perpendicular to the cell edge, red.

The online version of this article includes the following source data and figure supplement(s) for figure 2:

**Source data 1.** Microtubule (MT) directionality source data: KIF5B depletion and KIF5B KO.

**Figure supplement 1.** Microtubule (MT) abundance and alignment at the cell periphery depend on KIF5B in primary β cells in mouse islets.

**Figure supplement 2.** Overexpression of truncated KIF5B motor lacking cargo binding does not affect microtubule (MT) density at β-cell periphery.

**Figure supplement 3.** Workflow of microtubule (MT) directionality analysis.

**Figure supplement 4.** Illustration of thresholding variations and their influence on the output analysis.

to bind MTs as cargos. Neither in KIF5B-depleted nor in wt MIN6 cells, peripheral MTs were affected by the motor re-expression (***Figure 2—figure supplement 2***), indicating that kinesin-dependent MT growth does not noticeably contribute to the formation of the peripheral MT array in this cell type.

## KIF5B is required for β-cell sub-membrane MT array alignment

Given the known significance of the peripheral MT array, which normally consists of well-organized MTs parallel to the cell membrane (***Bracey et al., 2020***), we have further analyzed directionality of MTs remaining at the cell periphery after KIF5B depletion. Previously, we published a custom image analysis algorithm (***Bracey et al., 2020***) allowing for detailed quantitative characterization of MTs directionality in relation to the nearest cell border (***Figure 2—figure supplement 3***). Here, we applied the same computational analysis to MT imaging data in MIN6 cells with perturbed KIF5B level and/ or function. After deconvolution for increased signal-to-noise ratio, single 2D slices of MT images were subjected to thresholding. Two thresholding options were analyzed for the unbiased approach. A threshold specifically optimized (perfect) for the peripheral MT array in each cell was considered along with a standard threshold across the analyzed cell population (***Figure 2—figure supplement 4***). Thereafter, the directionality of MTs with respect to the cell border was determined. Every pixel of the image was analyzed with inconclusive pixels disregarded. MT directionality was quantified as a function of the distance from the cell border and directionality of peripheral MTs within 1 µm of the cell border quantified. Across the analysis, two thresholding conditions provided similar outputs. Our results indicate that in MIN6 cells treated with nontargeting control shRNA (***Figure 2F***), the distribution of MT angles in the cell periphery is vastly parallel and co-aligned with the cell boundary, as previously reported for primary islet β cells (***Bracey et al., 2020***). In contrast, the loss of KIF5B via shRNA depletion resulted in a significant loss of parallel MTs at the periphery as compared to control (***Figure 2E–H***).

Furthermore, the MT directionality analysis in β cells within islets from the *Kif5b* KO mice revealed a significant decrease in MT alignment to the cell periphery as compared to the wt mice, closely mirroring our findings from the *Kif5b* knockdown in MIN6 cells (***Figure 2—figure supplement 1D–G***). Collectively, our data suggest that KIF5B-mediated MT sliding serves to facilitate MT transport to the cell periphery and align them at this cellular location in both a β-cell culture model and primary β cells within intact islets.

Combined, our data demonstrate a distinct effect of KIF5B perturbation on both the distribution of MTs to the cell periphery and their orientation along the cell boundary. These data suggest that KIF5B-driven MT sliding is a decisive mechanism of the sub-membrane MT array generation, likely via redistribution of centrally nucleated MTs and their subsequent alignment at the cell edge. Thus, MT sliding is likely a critical component in functional MT organization in β cells.

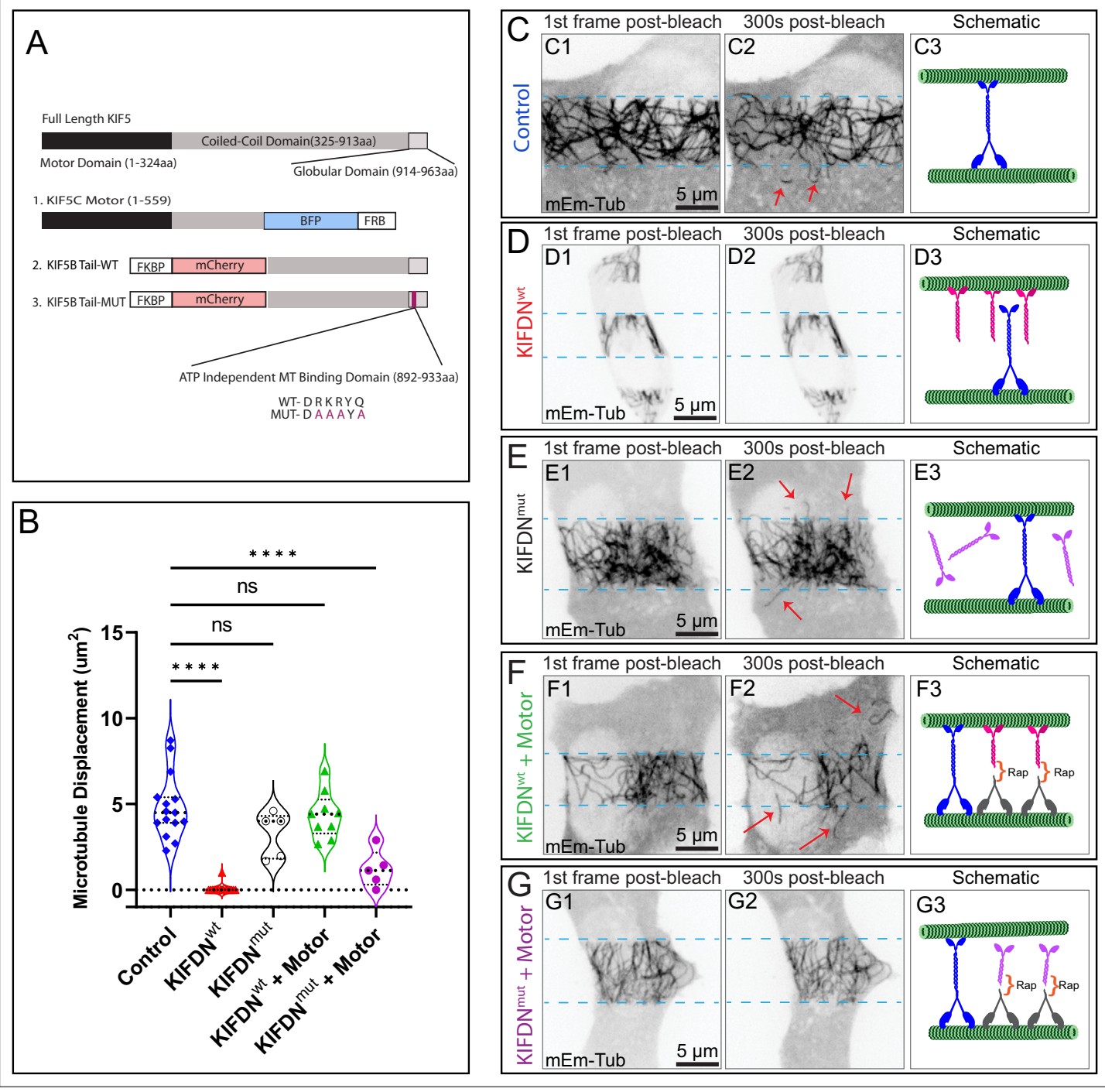

**Figure 3.** Microtubule (MT) sliding is facilitated through the ATP-independent MT-binding domain of kinesin-1. (**A**) Schematic of kinesin-1 (KIF5) and the dominant-negative (KIFDN) and heterodimerization strategy. Top schematic shows full-length KIF5s, consisting of the motor domain, stalk coil-coil domain, and the tail. Three constructs utilized here include (1) The KIF5C motor domain tagged with a blue fluorescent protein (BFP) and the FKBP-rapamycin binding (FRB) for heterodimerization; (2) KIFDN$^{wt}$ construct with KIF5B tail domain tagged with the mCherry fluorescent protein and the FKBP for heterodimerization. (3) KIFDN$^{mut}$ construct is the same as (2) but features a set of point mutations (magenta) making the ATP-independent MT-binding domain unable to bind MT lattice. (**B**) Quantification of MT sliding in FRAP assay in cells subjected to DN construct expression and heterodimerization. MT displacement is shown as the area of MTs displaced into the bleached area after 5 min of recovery. See representative data (**C–G**). N=5–25 per condition. One-way ANOVA test was performed for statistical significance (p-value<0.0001; ns, nonsignificant). (**C–G″**) Frames from representative FRAP live-cell imaging sequences of MIN6 cells expressing mEmerald-tubulin. Inverted grayscale images of maximum intensity projections over 1-μm-thick stacks by spinning disk confocal microscopy. (C1–G1) The first frame after photobleaching. (C2–G2) A frame 5 min (300 s) after photobleaching. Light-blue dotted lines indicate the edges of the photobleached areas. Red arrows indicate MTs displaced into the bleached area.

*Figure 3 continued on next page*

*Figure 3 continued*

Scale bars, 5 µm. (C3–G3) Schematics of experimental manipulation: green represents MTs, blue represents endogenous KIF5B, magenta represents KIFDN^wt, purple represents KIFDN^mut, gray represents KIF5C motor, orange bracket represents heterodimerizing agent (rap, rapalog). Conditions: (C1–C3) Untreated control. Only endogenous KIF5B is present. (D1–D3) KIFDN^wt overexpression. Endogenous KIF5B is unable to bind MTs. (E1–E3) KIFDN^mut overexpression. It does not bind MTs and does not interfere with endogenous KIF5B. (**C–E**, *Figure 3—video 1* 'KFDN FRAP') (F1–F3) KIFDN^wt and KIF5C motor overexpression plus rapalog treatment. Heterodimerization creates a large pool of motors capable of MT sliding. (**G–G"**) KIFDN^mut and KIF5C motor overexpression plus rapalog treatment. Heterodimerization creates a large pool of the motor nonfunctional in MT sliding (**F–G**, *Figure 3—video 2* 'KFDN + motor FRAP').

The online version of this article includes the following video(s) for figure 3:

**Figure 3—video 1.** KFDN FRAP.

https://elifesciences.org/articles/89596/figures#fig3video1

**Figure 3—video 2.** KFDN + motor FRAP.

https://elifesciences.org/articles/89596/figures#fig3video2

## β-Cell kinesin-1 drives MT sliding through the C-terminal MT-binding domain

While membrane cargo transport by KIF5s requires association of the heavy chain with the kinesin light chains (KLCs) and/or other adaptors, transportation of MTs as cargos occurs due to direct binding of KIF5 to MTs through the ATP-independent MT binding domain in heavy chain tail (C-terminus) (*Jolly et al., 2010*; *Seeger and Rice, 2010*).

To specifically establish the role of MT sliding by KIF5B in β cells, we sought to evaluate the effects of suppressing the binding of KIF5B tail to MTs. To this end, we used a previously generated construct (*Ravindran et al., 2017*), which is a motor-less version of wt kinesin-1 motor KIF5B containing the cargo-binding and ATP-independent MT binding domain and tagged with mCherry (mCh) at the amino terminus (*Figure 3A*). When overexpressed, this construct acts as a dominant-negative (DN) tool preventing the association of the tail of endogenous KIF5B with MTs. This tool is referred to as KIFDN^wt (KIF5B DN wt) moving forward (*Figure 3A*).

To confirm that the KIF5B tail domain binds to MTs in MIN6 and acts as a DN, we co-expressed KIFDN^wt and mEmerald-tubulin. When subjected to the FRAP assay, we detected a complete loss of MT sliding events as compared to a control (*Figure 3B–D*). To prevent tail engagement of the MT lattice through the ATP-independent binding domain, we opted to make point mutations in the tail domain to change the residues 892-DRKRYQ to 892-DAAAYA, thus generating KIFDN^mut (*Figure 3A*). Photobleaching assay in cells co-expressing the KIFDN^mut with mEmerald-tubulin indicated that the MT sliding activity was not blocked in the presence of the mutated construct (*Figure 3B and E*, *Figure 3—video 1* KFDN FRAP), confirming that KIF5B tail domain binding to MTs is needed for MT sliding in β cells.

The DN constructs are also tagged with the FK506-rapamycin-binding protein (FKBP), as indicated in *Figure 3A*. This allows to heterodimerize them with a motor domain fused with the FKBP-rapamycin binding (FRB) domain using A/C heterodimerizer (rapalog) and reconstitute a functional motor (*Inobe and Nukina, 2016*). We restored kinesin-1 activity by connecting the motor-less KIF5B, KIFDN^wt, to kinesin-1 motor domain as a way to rescue the effects of DN approach of KIF5B tail overexpression. To this end, we co-expressed MIN6 cells with the tail domain, mEmerald-tubulin, and the KIF5C motor domain fused to FRB domain (*Figure 3A*). Once the tail and motor domains were dimerized with rapalog, we saw that the once blocked MT sliding events of the KIFDN^wt tail alone were now reversed (*Figure 3B and F*, *Figure 3—video 2* KFDN + motor FRAP). In contrast, under conditions of heterodimerization of KIFDN^mut with the motor, MT sliding was greatly impaired (*Figure 3B and G*, *Figure 3—video 2* KFDN + motor FRAP), indicating that the motor with the mutated ATP-independent binding domain cannot use MTs as cargos. Interestingly, the endogenous motor in this case was unable to efficiently transport MTs, suggesting that the endogenous motor pool engaged in MT sliding was significantly smaller than the overexpressed nonfunctional motor.

Overall, the results of the DN approach confirm that MT sliding in β cells is driven by KIF5B through direct kinesin-1 tail binding to cargo MTs.

## Effects of C-terminal MT-binding of kinesin-1 on β-cell MT organization

Keeping in mind that KIF5B has additional major functions in addition to MT sliding, we sought to test the consequence of MT sliding more directly by turning to overexpression of the DN constructs. Thus, we took advantage of our heterodimerization approach to analyze MT patterns in cells with active kinesin-1, which is able or unable to slide MTs (see *Figure 3F* versus *Figure 3G*). We analyzed MIN6 cells that express either the KIFDN^wt or KIFDN^mut tail domains alone (*Figure 4—figure supplement 1*) or co-expressing and heterodimerized with the motor domain (*Figure 4B and C*). Cells were fixed and immunostained for tubulin to identify the MT network. As expected, overexpression of the KIFDN^wt tail construct alone acted as DN toward MT distribution to the cell periphery, resulting in decreased peripheral tubulin intensity (*Figure 4—figure supplement 1A and C*), while in cells expressing KIFD-N^mut, MT patterns were comparable to control (*Figure 4—figure supplement 1B and C*).

Interestingly, expression of heterodimerized kinesin motors led to impaired MT network configurations compared to NT control (*Figure 4*). Specifically, blocking of MT sliding by overexpression of KIFDN^mut heterodimerized with the motor led to the decrease in peripheral tubulin intensity (*Figure 4C and D*) and impaired MT aligning along the cell border (*Figure 4E and G*). These data indicate that KIF5B-driven relocation of centrally nucleated MTs to β-cell periphery requires kinesin tail domain binding to 'cargo' MTs. Strikingly, overexpression of functional heterodimerized motor, which was capable of MT sliding and populating of the cell periphery with MTs as detected by tubulin intensity readings (KIFDN^wt heterodimerized with the motor, *Figure 4B and D*), also led to a deficient MT aligning at the periphery (*Figure 4E and H*). This can be interpreted as a result of unregulated sliding in these experimental conditions, since excessive kinesin-1-dependent sliding can lead to MT bending (*Straube et al., 2006*). This suggests that proper organization of MTs within the sub-membrane array requires fine-tuning of MT sliding activity.

Collectively, this indicates that regulated KIF5B activity is essential for redistributing MTs to the cell border and sustaining an aligned peripheral MT array.

## Exaggerated MT sliding leads to defects in peripheral array alignment

Our data discussed above suggest that overexpression of functional kinesin-1 disrupts MT alignment at the cell periphery, inducing their bending and buckling (*Figure 4E and G*). To test if this defect is a result of excessive MT sliding, we employed a small molecule, kinesore, which is known to dramatically promote MT sliding by kinesin-1 (*Randall et al., 2017*). Kinesore targets kinesin cargo adaptor function by impairing KLC from binding kinesin heavy chain. As a result, kinesin heavy chain will excessively engage MTs through the C-terminal, ATP-independent MT-binding domain, leading to exaggerated MT sliding and the loss of membrane cargo transport by kinesin-1 (*Randall et al., 2017*). To this end, we pretreated MIN6 cells with 50 μm kinesore and stained for MTs (*Figure 5*).

We have observed an exaggerated MT looping resulting in a slight decrease in peripheral MT intensity (*Figure 5C*). To validate whether the effect of kinesore on MT morphology was KIF5B-dependent, we applied kinesore treatment to cells treated with either scrambled control or *Kif5b*-specific shRNA (*Figure 5—figure supplement 1*). MT organization in KIF5B-depleted cells was not affected by kinesore, indicating that the observed effect was likely due to kinesore-induced MT sliding exaggeration (*Figure 5—figure supplement 2*). Further analysis of the peripheral bundle indicated that MT alignment was strongly impaired upon kinesore-driven MT remodeling as compared with vehicle (DMSO) treatment (*Figure 5D–G*). The loss of co-aligned MTs and loss of tubulin density at the periphery indicate that MT sliding must be gated to prevent overcorrected MT networks.

## MT sliding in β cells is activated by glucose stimulation

It has previously been reported that kinesin-1 switches activity level in the presence of glucose stimuli (*Donelan et al., 2002*). We predicted that, as KIF5B activity modulates the MT sliding events, they would also change depending on the glucose concentration. To test this, we preincubated MIN6 cells with media containing a low concentration of 2.8 mM glucose (*Figure 6A*). We applied the photobleaching assay under these conditions and detected little to no MT sliding events. When switching glucose to a high concentration of 20 mM, MT sliding and remodeling events were significantly increased (*Figure 6B*, *Figure 6—video 1* 'FRAP low and high glucose'). Quantification of the sliding events demonstrated that MIN6 displaced MTs via MT sliding significantly more efficiently upon glucose stimulation (*Figure 6C*). We then turned to single-molecule tracking of MT lattice fiducial

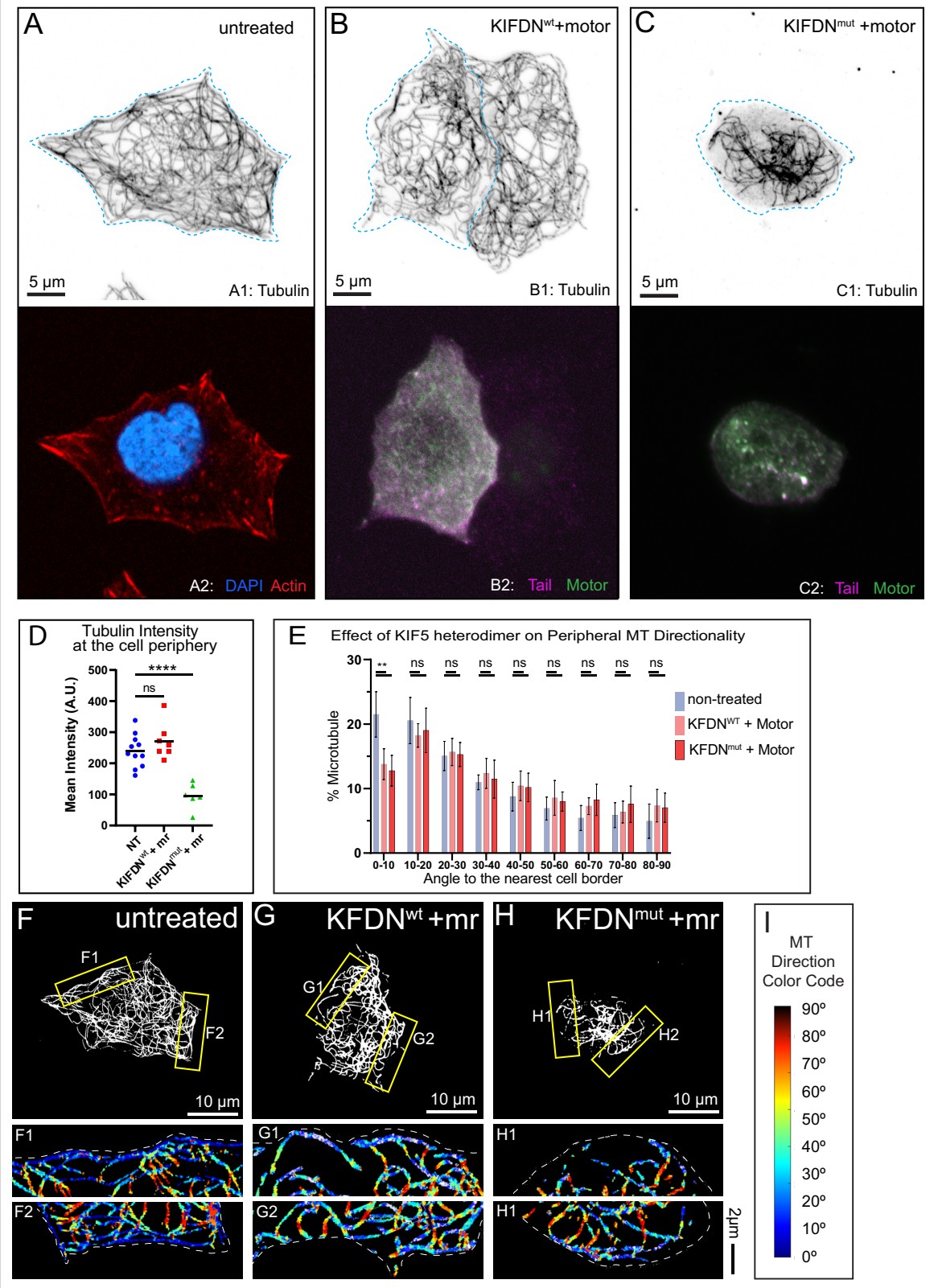

**Figure 4.** Effects of ATP-independent microtubule (MT)-binding domain of KIF5B on MT abundance and alignment at the β-cell periphery. (A–C) MT organization in MIN6 cells expressing (B) KIFDN^wt and KIF5C motor heterodimerized via rapalog treatment, (C) KIFDN^mut and KIF5C motor heterodimerized via rapalog treatment and compared to a control cell with no ectopic expressions (A). Top, immunofluorescence staining for tubulin (grayscale, inverted). Blue dotted line indicates the borders of a cell expressing constructs of interest. Bottom in **A**, F-actin (phalloidin, red) and DAPI

*Figure 4 continued on next page*

*Figure 4 continued*

(blue). Bottom in **B** and **C**, ectopically expressed mCherry-labeled KIFDN constructs (magenta) and BFP-labeled KIF5C motor (green). Laser scanning confocal microscopy maximum intensity projection of 1 μm at the ventral side of the cell. Scale bars, 5 um. (**D**) Quantification of mean tubulin intensity within the outer 2 μm peripheral area of a cell, in data represented in (**A–C**). Mean values, black bars. One-way ANOVA, $p<0.0001$. N=7–15 cells. (**E**) Histograms of MT directionality within 1 μm of cell boundary (see *Figure 2—figure supplement 3* for the analysis workflow) in control cells compared to cells expressing heterodimerized KIFDN variants. Data are shown for the summarized detectable tubulin-positive pixels in the analyzed cell population, as represented in (**F–H**). Unpaired t-tests were performed across each bin for all cells, and a Kolmogorov-Smirnov (K-S) test was performed on the overall distribution. The share of MTs parallel to the edge (bin 0–10) is significantly higher in control as compared to the overexpressions. NT control N=138,810 pixels across 9 cells, KIFDN^wt + motor N=48,285 pixels across 9 cells, KIFDN^mut + motor N=40,832 pixels across 10 cells. (**F–H**) Representative examples of MT directionality analysis quantified in (**E**). (**F**) Control cell, no ectopic expressions. (**G**) Cell expressing KIFDN^wt + motor. (**H**) Cell expressing KIFDN^mut+ motor. Overviews of cellular MT networks are shown as a threshold to detect individual peripheral MTs (see *Figure 2—figure supplement 3* panel A5). (F1–H2) Directionality analysis outputs of regions from yellow boxes in (**F–H**) are shown color-coded for the angles between MTs and the nearest cell border (see *Figure 2—figure supplement 3* panel A8). (**I**) Color code for (F1–H2): MTs parallel to the cell edge, blue; MTs perpendicular to the cell edge, red.

The online version of this article includes the following source data and figure supplement(s) for figure 4:

**Source data 1.** Microtubule (MT) directionality source data: KIFDN overexpression.

**Figure supplement 1.** Dominant-negative (DN) effect of KIF5B tail domain on microtubule (MT) abundance at the β-cell periphery requires ATP-independent MT-binding domain.

**Figure supplement 2.** Influence of thresholding variations on the output analysis of microtubule (MT) directionality in *Figure 4F–H*.

marks (K560Rigor^E236A-SunTag) to further investigate this observation. Consistent with the photo-bleaching assay, the fiducial marks were predominantly stationary in cells preincubated in 2.8 mM glucose (*Figure 6D*) but frequently underwent directed relocation events indicative of MT sliding in cells after being stimulated with 20 mM glucose (*Figure 6E*, *Figure 6—video 2* 'SunTag low and high glucose').

These data demonstrate that glucose-stimulated remodeling of the MT network involves regulated MT sliding. Given the importance of MT sliding for peripheral MT abundance (*Figure 7A*) and alignment (*Figure 7B*), this effect may be essential to remodel parts of the peripheral MT array during GSIS to allow for secretion (*Figure 7C*) and/or to restore the initial array after glucose-dependent destabilization (*Figure 7D*).

## Discussion

Since the first description of the convoluted MT network in MIN6 cells by the Rutter group (*Varadi et al., 2003*), our views on the regulation, function, and dynamics of the pancreatic β-cell MT network have been gradually evolving (*Bracey et al., 2022*). However, the field is still far from understanding the mechanisms underlying the network architecture. Here, we show that MT sliding is a prominent phenomenon in β cells and that it is driven by kinesin KIF5 B. This kinesin-1-dependent MT sliding is a critical mechanism needed for the formation and long-term maintenance of β-cell MT networks. In addition, we show that MT sliding activity is facilitated by glucose stimulation, suggesting that this process is involved in the regulation of GSIS and/or providing β-cell fitness during the response to glucose. Overall, our study establishes MT sliding as an essential regulator of β-cell architecture and function.

As we reviewed in the Introduction, the β-cell MT network consists of an interlocked meshwork and peripheral MT arrays co-aligned with the cell border. We find that blocking kinesin has two major effects on MT organization: (1) receding of MTs at the cell periphery and increased MT density in the cell center, and (2) lack of alignment of remaining peripheral MTs with the cell border. We interpret that the first phenotype arises from the lack of MT sliding from the cell center to the periphery, and the second phenotype arises from the lack of MT sliding along pre-existing peripheral MTs (*Figure 7*).

In cultured mesenchymal cells, the kinesin motor was reported to populate the cell periphery with MTs via promoting rescues and elongation of radially arranged dynamic MTs (*Andreu-Carbó et al., 2022*). Our data indicate that this mechanism does not noticeably contribute to kinesin-dependent MT organization in β cells, leaving MT sliding the only known mechanism underlying the phenotype observed here. This difference in kinesin action could be due to basic differences in MT network

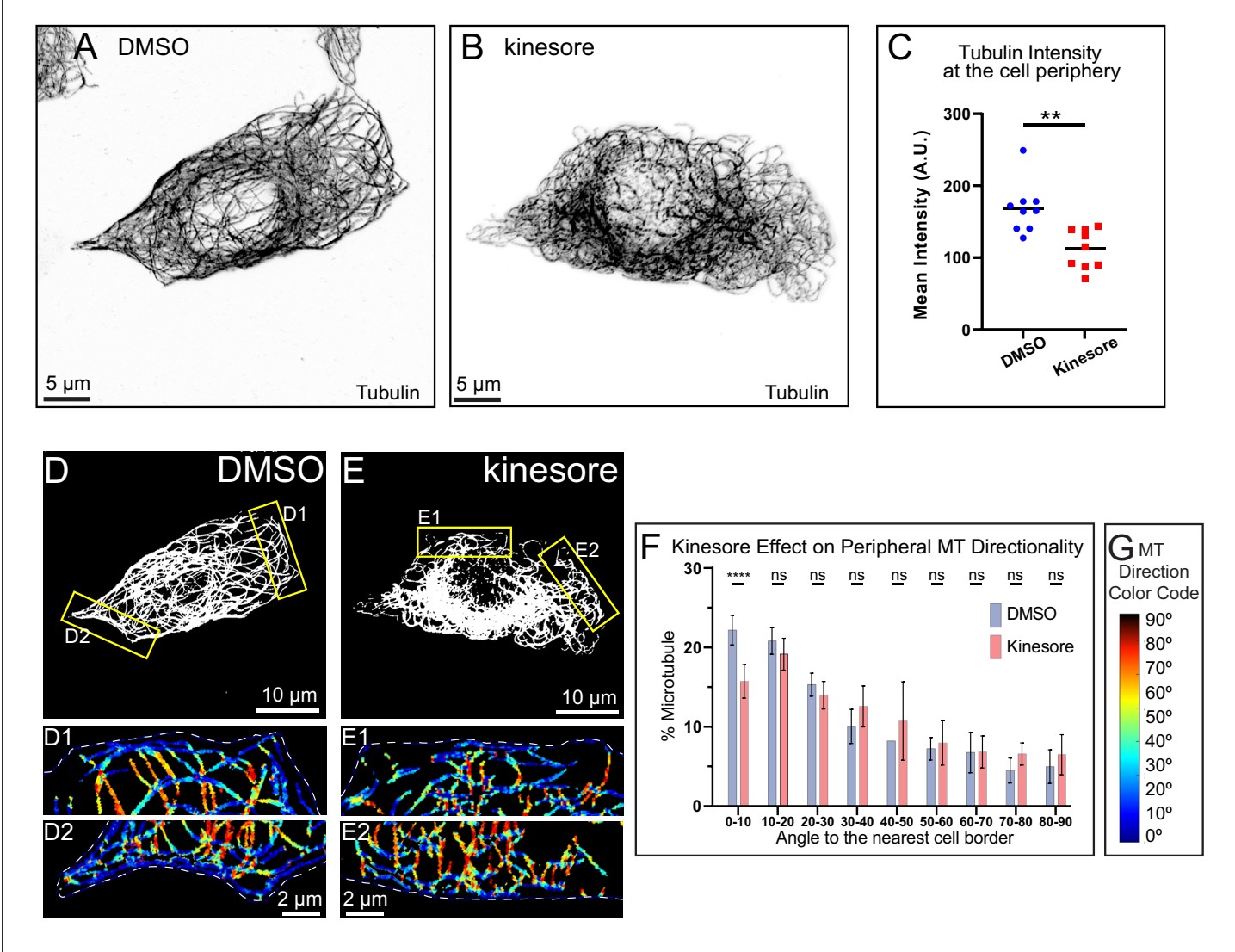

**Figure 5.** Enhanced microtubule (MT) sliding results in loss of peripheral MT alignment at the border. (**A–B**) MT organization in MIN6 cells pretreated with DMSO and kinesore, respectively. Immunofluorescence staining for tubulin (grayscale, inverted). Laser scanning confocal microscopy maximum intensity projection of 1 μm at the ventral side of the cell. Scale bars, 5 um. (**C**) Quantification of mean tubulin intensity within the outer 2 μm peripheral area of a cell, in data represented in (**A–B**). Mean values, black bars. One-way ANOVA, p<0.0001. N=10 cells per condition. Histograms of MT directionality within 1 μm of cell boundary (see *Figure 2—figure supplement 3* for the analysis workflow) in DMSO-treated control cells compared to kinesore-treated cells. Data are shown for the summarized detectable tubulin-positive pixels in the analyzed cell population, as represented in (**D–E**). Unpaired t-tests were performed across each bin for all cells, and a Kolmogorov-Smirnov (K-S) test was performed on the overall distribution. The share of MTs parallel to the edge (bin 0–10) is significantly higher in control as compared to the overexpressions. DMSO control N=136,840 pixels across 10 cells, kinesore-treated N=87,361 pixels across 9 cells. (**D–E**) Representative examples of MT directionality analysis quantified in (**F**). Directionality analysis outputs of regions from yellow boxes in (**D–E**) are shown color-coded for the angles between MTs and the nearest cell border (see *Figure 2—figure supplement 3* panel A8). (**G**) Color code for (D1–E2): MTs parallel to the cell edge, blue; MTs perpendicular to the cell edge, red.

The online version of this article includes the following source data and figure supplement(s) for figure 5:

**Source data 1.** Microtubule (MT) directionality source data: kinesore treatment.

**Figure supplement 1.** Influence of thresholding variations on the output analysis of microtubule (MT) directionality in *Figure 5D and E*.

**Figure supplement 2.** Kinesore has no effect in microtubule (MT) network in KIF5B-depleted cells.

properties in these cell types: in a mostly non-radial, highly stabilized MT network in β cells (*Zhu et al., 2015*), modulation of MT plus end rescue efficiency is not likely to be a significant factor.

Interestingly, blocking kinesin results in a striking accumulation of MT in the cell center where they are normally nucleated at MTOCs, which include the centrosome and the Golgi, in differentiated β

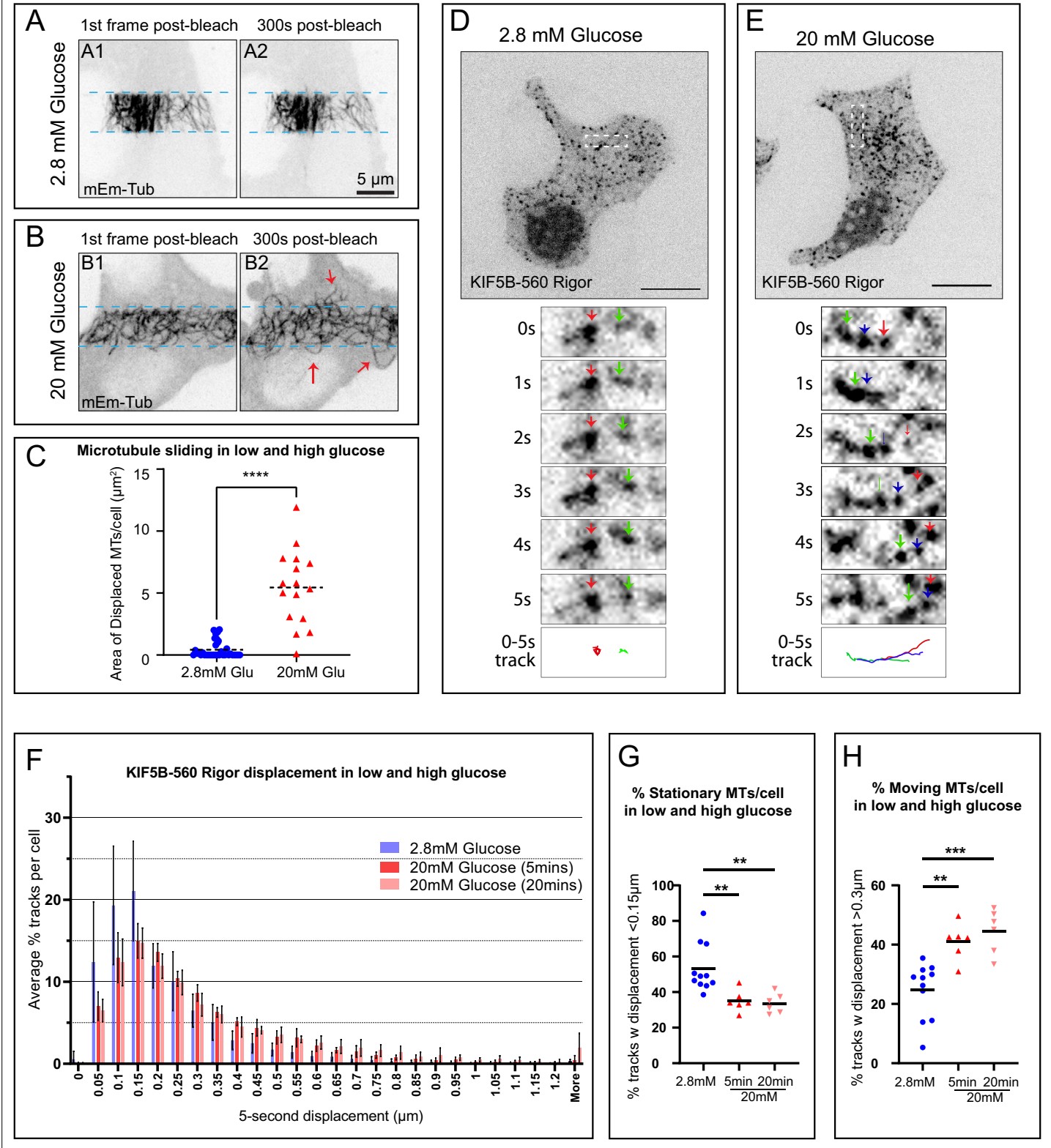

**Figure 6.** Microtubule (MT) sliding in β cells is stimulated by glucose. (**A–B**) Frames from representative FRAP live-cell imaging sequences of MT sliding response to glucose stimulation. mEmerald-tubulin-expressing MIN6 cells. Inverted grayscale images of maximum intensity projections over 1-μm-thick stacks by spinning disk confocal microscopy. (**A**) A cell pretreated with 2.8 mM glucose before the assay. (**B**) A cell pretreated with 2.8 mM glucose and stimulated with 20 mM glucose before the assay. (A1–B1) The first frame after photobleaching. (A2–B2) A frame 5 min (300 s) after photobleaching. Light-blue dotted lines indicate the edges of the photobleached areas. Red arrows indicate MTs displaced into the bleached area. Scale bars, 5 μm.

*Figure 6 continued on next page*

*Figure 6 continued*

(**C**) Quantification of MT sliding FRAP assay in cells in 2.8 mM versus 20 mM glucose (see representative data in **A**–**B**). MT displacement is shown as the area of MTs displaced into the bleached area after 5 min of recovery. One-way ANOVA test was performed for statistical significance (p-value<0.0001). N=16–24 cells per set (**A**–**B**, *Figure 6—video 1* 'FRAP low and high glucose'). (**D**–**E**) MIN6 cells featuring fiducial marks at MTs due to co-expression of SunTag-KIF5B-560Rigor construct and Halo-SunTag ligand. Representative examples for cells in 2.8 mM glucose (**D**) and a cell stimulated by 20 mM glucose (**E**) are shown. Single-slice spinning disk confocal microscopy. Halo-tag signal is shown as inverted grayscale image. Top panels show cell overviews (scale bars, 5 µm). Below, boxed insets are enlarged to show dynamics of fiducial marks (color arrows) at 1 s intervals (1–5 s). 0–5 s tracks of fiducial mark movement are shown in the bottom panel, each track color-coded corresponding to the arrows in the image sequences. N=6–11 cells (**A**–**B** *Figure 6—video 2* 'SunTag low and high glucose'). (**F**) Histogram of all 5 s displacement of fiducial marks in low versus high glucose. (**G**) Summarized quantification of stationary fiducial marks along MT lattice (5 s displacements below 0.15 µm). Low glucose N=5615 tracks across 11 cells, high glucose 5 min N=2259 tracks across 6 cells, high glucose 20 min N=3059 tracks across 6 cells. One-way ANOVA, p<0.001. (**H**) Summarized quantification of moving fiducial marks along the MT lattice (5 s displacements over 0.3 µm). Low glucose N=2595 tracks across 11 cells, high glucose 5 min N=2642 tracks across 6 cells, high glucose 20 min N=4049 tracks across 6 cells. One-way ANOVA, p<0.001.

The online version of this article includes the following video and source data for figure 6:

**Source data 1.** SunTag marks displacement 5 s intervals across each cell and population analysis referenced in panels G and H.

**Figure 6—video 1.** FRAP low and high glucose.

https://elifesciences.org/articles/89596/figures#fig6video1

**Figure 6—video 2.** SunTag low and high glucose.

https://elifesciences.org/articles/89596/figures#fig6video2

cells, the latter being the main MTOC. Thus, sliding MTs originate from the MTOC area. At the same time, FIB-SEM analysis did not detect many MTs associated with MTOCs in physiologically normal β cells (*Müller et al., 2021*). This implies that MTs are typically rapidly dissociated from MTOCs so that they become available for transport by sliding. It is worth mentioning that for long-distance transport by sliding, cargo MTs must be short; otherwise, MT buckling and not long-distance transport will occur (*Straube et al., 2006*). Interestingly, shorter MTs have been observed in high glucose conditions (*Müller et al., 2021*) when MTs are nucleated more actively (*Trogden et al., 2019*) and transported more frequently (this paper). Possibly, nucleated MTs are detached from MTOCs before they achieve a length that would prevent their transport. An intriguing possibility has been proposed, indicating that in high glucose conditions, MTs might undergo severing by katanin (*Müller et al., 2021*). This process could generate MT fragments, potentially facilitating their role as cargos with increased ease of transport. It is also possible that sliding MT subpopulation has some additional specific features that make them preferred cargos, since it is becoming increasingly clearer in the field that there is immense heterogeneity among MTs. Posttranslational modifications and MT-associated proteins, which vastly alter stability and coordination of motor proteins (*Hammond et al., 2008*; *McKenney et al., 2016*; *Monroy et al., 2018*; *Yu et al., 2015*), might also influence which MTs serve as cargos versus transportation tracks in β cells.

Importantly, we observe a prominent effect of peripheral MT loss only after a long-term kinesin depletion (3–4 days). This is consistent with our observation that only a minor subset of MT is being moved within each experimental time frame. We postulate that the absence of a peripheral MT array in KIF5B-depleted cells is a consequence of prolonged lack of sliding. We hypothesize that MT sliding must contribute to β-cell-specific peripheral MT bundle formation during β-cell differentiation. We also found that increasing MT sliding does not yield a properly configured MT array: kinesore-treated cells lack aligned peripheral MTs, consistent with kinesore-induced MT looping reported in other cell types (*Randall et al., 2017*). This indicates that, similar to other parts of β-cell physiology, the dose of MT sliding has to be precisely tuned to achieve physiologically relevant architecture. It was shown before that exaggerated kinesin-dependent MT sliding causes MT bundling and buckling into aberrant configuration (*Straube et al., 2006*). We predict that a fine-tuning regulatory pathway must exist to restrict the number of MT sliding events to the cell needs.

Consistent with this idea, MT sliding is sensitive to metabolic regulation: our data indicate that MT sliding is activated on a short-term basis after glucose stimulation. While the deep understanding of the role of MT sliding in GSIS requires further studies, it is plausible to suggest two potential functions for this process in glucose-stimulated cells. (1) Given that peripheral MTs are destabilized in high glucose (*Ho et al., 2020*), we suggest that a long-term function for MT sliding is needed to replace MT population at the cell periphery and restore the pre-stimulus MT organization (*Figure 7B*).

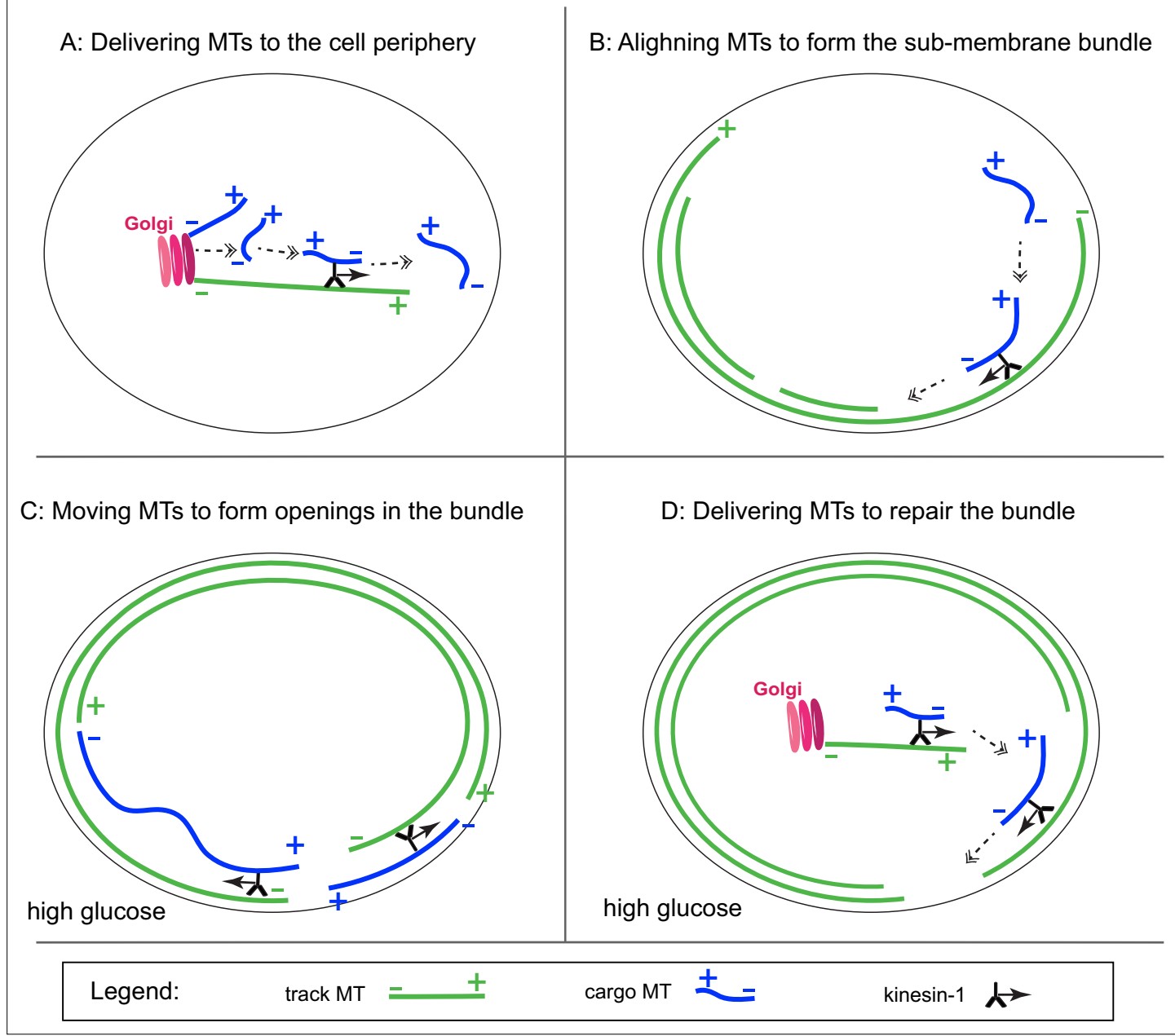

**Figure 7.** Schematic of the main results and predictions. (**A**) Role of KIF5B-dependent microtubule (MT) sliding in MT density at the cell periphery. (**B**) Role of KIF5B-dependent MT sliding in the alignment of peripheral MTs. (**C**) Potential role of glucose-facilitated KIF5B-dependent MT sliding in forming temporal openings in the peripheral MT array by moving fragmented MTs aside or MT looping. (**D**) Potential role of glucose-facilitated KIF5B-dependent MT sliding in repair of the peripheral MT array, partially destabilized/fragmented downstream of glucose. Track MTs, green. Cargo MTs, blue. Kinesin-1, black. Direction of kinesin movement shown as a solid arrow. Subsequent steps of the process are shown as dashed double arrows.

Because the amount of MT polymer on every glucose stimulation changes only slightly (*Müller et al., 2021*; *Zhu et al., 2015*) and we detect MT loss from the periphery only after a prolonged blocking of sliding, we reason that this function could be essential to maintain long-term β-cell fitness and prepare cells for repeated rounds of stimulation. (2) As a potential short-term function at each stimulation round, MT sliding within the peripheral bundle itself could rearrange fragmented MTs within this array suppressing its role in IG withdrawal from specific secretion sites (*Figure 7B*). In this scenario, MT sliding would tilt the balance between positive and negative MT-dependent regulation of GSIS toward enhanced secretion at each stimulation.

Our finding provides an example of a phenomenon where a slight change in MT configuration is physiologically significant. This is not an exception because subtle MT defects often have dramatic consequences. For example, in mitotic spindles, a tiny overgrowth of MT ends during metaphase, which causes them to attach to both kinetochores rather than just one, is very significant for the efficiency of chromosome segregation, causing aneuploidy and cancer. The changes in β-cell MT networks that we are reporting are much stronger: the effect on the peripheral MT network accumulated over 3 days of KIF5B depletion is dramatic (*Figure 2B and C*). Short-term gross MT network configurations after a single glucose stimulation are harder to detect but are consistent with previous reports that MTs at the cell periphery are destabilized and fragmented upon high glucose stimulus (*Ho et al., 2020*; *Müller et al., 2021*), and that preventing this MT rearrangement completely blocks GSIS (*Zhu et al., 2015*; *Ho et al., 2020*).

One of the most fascinating features of insulin secretion regulation is that the amount of generated insulin granules significantly exceeds the normal physiological needs for insulin secretion (~100 times more than needed). At the same time, even slightly facilitated glucose depletion can be devastating for the human body. Accordingly, the excessive insulin content of a β cell resulted in the development of multiple levels of control, preventing excessive secretion. Our previous data suggest that the peripheral MT array provides one of those mechanisms. This study indicates that MT sliding is necessary to form the proper peripheral network in the long term. Short-term glucose-induced changes in the peripheral MT array likely need to be subtle to prevent oversecretion. Thus, we are not surprised that a dramatic effect of sliding inhibition is only detectable by our approaches after the changes in the MT network accumulate over time.

On a final note, it is important to evaluate the phenomenon reported here in light of the dual role of KIF5B as IG transporter and MT transporter and the coordination of those two roles in IG transport and availability for secretion. Our results indicate that KIF5B is needed for the formation of the peripheral MT bundle, which we have shown to restrict secretion (*Bracey et al., 2020*; *Ho et al., 2020*). At the same time, it is well established that KIF5B transports IGs, and KIF5B loss of function impairs GSIS (*Meng et al., 1997*, *Varadi et al., 2002*; *Cui et al., 2011*). After a prolonged KIF5B inactivation, a loss of peripheral readily releasable IG should be expected due to two factors: because there is no MT bundle to prevent oversecretion and IG depletion, and because there is no new IGs being transported from the Golgi area. In contrast, physiological activation of kinesin by glucose (*Donelan et al., 2002*; *Varadi et al., 2003*) would both promote replenishment of IG through nondirectional transport through the cytoplasm and restoration of the peripheral MT array to prevent oversecretion.

In conclusion, here, we add another very important cell type to the list of systems that employ KIF5B-dependent MT sliding to build functional cell-type-specific MT networks. This system is unique because, in this case, MT sliding is metabolically regulated and activated on a single-minute timescale by nutrition triggers.

# Materials and methods
## Key reagents
**Key resources table**

| Reagent type (species) or resource | Designation | Source or reference | Identifiers | Additional information |
|---|---|---|---|---|
| Cell line (*M. musculus*) | MIN6 | Gift from Dr. Miyazaki, Osaka University Medical School, Japan | RRID:CVCL_0431 | Mouse Insulinoma Cell Line |
| Chemical compound, drug | Kinesore | Tocris Bioscience | Cat#: 6664 | Final concentration (50 µm) |
| Chemical compound, drug | A/C Heterodimerizing Drug | Takara Bio Inc | Cat#: 635056 | Final concentration (25 µm) |

*Continued on next page*

*Continued*

| Reagent type (species) or resource | Designation | Source or reference | Identifiers | Additional information |
|---|---|---|---|---|
| Antibody | Anti-KIF5B rabbit monoclonal antibody | Abcam | Cat#: Ab167429 | (1:500 dilution) |
| Antibody | Anti-alpha-Tubulin mouse monoclonal antibody (DM1a) | Sigma-Aldrich | Cat#: T9026 | (1:1000 dilution) |
| Antibody | Anti-alpha-Tubulin rabbit polyclonal antibody | Sigma-Aldrich | Cat#: T3526 | (1:1000 dilution) |
| Other | Janelia Fluor 585 (JF585) and 646 (JF646), HaloTag Ligands | Promega | CS315105, GA1120 | Final concentration (2 µl/mL) |
| Strain, strain background (*M. musculus*) | C57BL/6J | Jackson Laboratory | Strain #:000664 RRID:IMSR_JAX:000664 | Control mice (wt) |
| Strain, strain background (*M. musculus*) | *Kif5b*$^{fl/-}$:*RIP2-Cre* | *Cui et al., 2011* | Strain #:008637; RRID:IMSR_JAX:008637 | *Kif5b* KO mice; derived from Jackson Laboratory fl/fl strain (see identifiers) |

## Cell lines

MIN6 cells (a gift from Dr. Miyazaki, Osaka University Medical School, Japan) between passages 40 and 60 were utilized (*Ishihara et al., 1993*; *Miyazaki et al., 1990*). Cells were authenticated on arrival (passage 40) for murine origin by STR profiling. Thereafter, β-cell identity was routinely confirmed by insulin content per cell (by immunostaining) and responsiveness to glucose (by standard GSIS ELISA, Alpco Diagnostics [Salem, NH], cat #80-INSMSU-E10). Cells were also routinely tested for mycoplasma by MycoFluor (ABD Biosciences, Rockville, MD, USA). Cells were maintained in 25 mM glucose Dulbecco's Modified Eagle Medium (DMEM) (Life Technologies, Frederick, MD, USA) supplemented with 10% fetal bovine serum (FBS), 0.001% β-mercaptoethanol, 0.1 mg/mL penicillin, and 0.1 mg/mL streptomycin in 5% $CO_2$ at 37°C.

## Mice

Mouse usage followed protocols approved by the Vanderbilt University Institutional Animal Care and Use Committee for GG and IK (protocol #M18000195). Mice were euthanized by isoflurane inhalation. C57BL/6J (Strain #:008637; RRID:IMSR_JAX:008637) mice were from The Jackson Laboratory (Bar Harbor, ME, USA). Conditional knockout *Kif5b*$^{fl/-}$:*RIP2-Cre* were bred and genotyped as previously described (*Cui et al., 2011*).

## Islet isolation and cell/islet culture

Islets were isolated from 8- to 16-week-old mice. Briefly, ~2 mL of 0.8 mg/mL collagenase P (MilliporeSigma, St. Louis, MI, USA) in Hanks' balanced salt solution (Corning, Corning, NY, USA) was injected into the pancreas through the common bile duct. The pancreas was digested at 37°C for 20 min. Islets were handpicked into RPMI 1640 media with 11 mmol/L glucose (Gibco, Dublin, Ireland) plus 10% heat-inactivated (HI) FBS (Atlanta Biologicals, Flowery Branch, GA, USA) and cultured at 37°C with 5% $CO_2$. For MIN6 cells, DMEM with 25 mmol/L glucose, 0.071 mmol/L β-mercaptoethanol (MilliporeSigma), 10% HI FBS, 100 µU/mL penicillin, and 100 µg/mL streptomycin (Gibco) was used.

## Reagents and antibodies

Primary antibodies for immunofluorescence were: mouse anti-β-tubulin (Sigma-Aldrich, 1:1000), rabbit anti-β-tubulin (Sigma-Aldrich, 1:1000), and rabbit anti-KIF5B (Abcam), Alexa488-, Alexa568-, and Alexa647-conjugated highly cross-absorbed secondary antibodies (Invitrogen). Coverslips were mounted in Vectashield Mounting Medium (Vector Laboratories). Cells were treated with indicated drugs for 3 hr unless otherwise indicated. Drugs used were: kinesore (Tocris Bioscience).

## shRNA sequence

The *Kif5b*-targeting shRNA [shRNA *Kif5b*] #1, [TL510740B, 5'-ACTCTACGGAACACTATTCAGTGG CTGGA] and [shRNA *Kif5b*] #2, [TL51074CB 5' – AGACCGTAAACGCTATCAGCAAGAAGTAG] are in the plasmid backbone pGFP-C-shLenti and were from Origene (Rockville, MD, USA). The nontargeting shRNA control was pGFP-C-shLenti, also from Origene.

## DNA constructs

| Plasmid construct | Source | Catalog # |
| --- | --- | --- |
| SCR-TGFP | Origene | Custom |
| *Kif5b* shRNA#1-TGFP | Origene | Custom |
| *Kif5b* shRNA#2-TGFP | Origene | Custom |
| Scr shRNA mEmerald-tubulin | This paper | Custom |
| *Kif5b* shRNA#1-mEmerald-tubulin | This paper | Custom |
| *Kif5b* shRNA#2-mEmerald-tubulin | This paper | Custom |
| mEmerald-tubulin-C-18 | Addgene | #54292 |
| pcDNA4TO-K560-E236A-24xGCN4 | Addgene | #60909 |
| ScFv-GCN4-HaloTag-GB1-NLS | Addgene | #106303 |
| FKBP-mCherry-KIF5B(568-964) | Kristen Verhey | *Ravindran et al., 2017* |
| p205ME_RnKIF5C(1-559)-TagBFP-FRB | Kristen Verhey | *Ravindran et al., 2017* |
| FKBP-mCherry-KIF5B(568-964)-AAAYA (MUT) | This paper | |

## Cloning

The Scr shRNA-mEmerald-tubulin, *Kif5b* shRNA#1-mEmerald-tubulin, *Kif5b* shRNA#2-mEmerald-tubulin were all generated from their respective TGFP containing constructs. Using the NotI and PmeI sites, the TGFP was swapped for mEmerald-tubulin-C-18.

The FKBP-mCherry-KIF5B(568-964) construct (gift from Kristen Verhey, University of Michigan) has previously been described (*Ravindran et al., 2017*).

By using site-directed mutagenesis, we made eight point mutations in the tail domain to change residues RKRYQ to AAAYA in the ATP-independent MT-binding domain. The point mutations were sufficient to rescue MT sliding in the cell. As previously published, this disrupts the tail domain to bind to the acidic e-hook of the MT tail. Point mutations were introduced using a site-directed mutagenesis kit, In-Fusion Snap Assembly (Takara).

## Lentiviral transduction and transfection

Lentivirus production and infection followed standard methods (*Huang et al., 2018*). MIN6 cells were treated with a given shRNA expressing an mEmerald-tubulin/cytosolic eGFP marker for 96 hr prior to imaging to achieve KD efficiency. For nonviral vectors, MIN6 cells were transfected using Amaxa Nucleofection (Lonza), and experiments were conducted 24–48 hr thereafter.

## Image acquisition

### Immunofluorescence microscopy of fixed samples

Fixed samples were imaged using a laser scanning confocal microscope Nikon A1r based on a TiE Motorized Inverted Microscope using a 100X lens, NA 1.49, run by NIS Elements C software. Cells were imaged in 0.05 μm slices through the whole cell.

### Live-cell imaging

Cells were cultured on four-chamber MatTek dishes coated with 10 μg/μL fibronectin and transduced 96 hr or transfected 48 hr before experiment. For live-cell imaging of MT sliding, cells were transfected with Emerald-tubulin and imaged using a Nikon TiE inverted microscope equipped with 488 and 568 nm lasers, a Yokogawa CSU-X1 spinning disk head, a PLAN APO VC 100x NA1.4 oil lens,

intermediate magnification ×1.5, and CMOS camera (Photometrics Prime 95B), 405 Burker mini-scanner, all controlled by Nikon Elements software.

## Photobleaching assay

~1 × 10⁶ MIN6 cells were transfected with 1 µg of mEmerald-tubulin or transduced with lentiviral KIF5B shRNA with mEmerald-tubulin as a reporter and attached to glass dishes coated with fibronectin for up to 96 hr. On the SDC microscope, the ROI tool in NIKON elements was used to place two ROIs~5 µm apart at either end of the cell. These regions were assigned to be photobleached with the equipped 405 nm mini-scanner laser leaving a fluorescent patch over the middle which we termed the 'fluorescent belt'. After the regions were photobleached, cells were then acquired for 5 min, across seven optical slices (0.4 µm step size) in 10 s interval between frames.

## SunTag rigor kinesin and tracking of MT sliding

SunTag system for MT lattice fiducial marks was adapted from *Lu et al., 2016*. ~1 × 10⁶ MIN6 cells were co-transfected with 1 µg of the ScFv-GCN4-HaloTag-GB1-NLS, and 0.5 µg of the pcDNA4TO-K560-E236A-24xGCN4 plasmid (K560Rigor^E236A-SunTag; *Tanenbaum et al., 2014*). After 24 hr, the cells were washed with 1x PBS and the media replaced with KRB containing 2.8 mM glucose for 1 hr, following a second incubation with HALO dye of choice (Promega) for 30 min. Cells were imaged in one focal plane for 2 min with 100 ms exposure time and no delay in acquisition.

## Image processing and presentation

*Figure 1*: (D–F) Maximum intensity projections of spinning disc confocal stacks through the ventral 1.2 µm of the cell for each time point. (G–I) Single focal plane spinning disc confocal time frames.

*Figure 1—figure supplement 1*: (A–C) Single focal planes from laser scanning confocal stacks.

*Figure 2*: (A–C) Maximum intensity projections of laser scanning confocal stacks over the ventral 1 µm of the cell. The tubulin channel (top) is shown as inverted grayscale images. The KIF5B channel (bottom) was pseudo-colored cyan. (F–H) Thresholded single focal plane images under the nucleus of the cell selected for directionality analysis.

*Figure 2—figure supplement 1*: (A, B) Maximum intensity projections of laser scanning confocal stacks over the ventral 1 µm of the islet. Inverted grayscale images. (D, E) Thresholded single focal plane images under the nucleus of the cell selected for directionality analysis.

*Figure 2—figure supplement 2*: (A–D) Maximum intensity projections of laser scanning confocal stacks through the whole cell. The tubulin channel (top) is shown as inverted grayscale images. The BFP-motor channel (bottom) was pseudo-colored cyan.

*Figure 2—figure supplement 3*: Single-plane laser confocal images shown as inverted grayscale images (A1–A2), thresholded images (A4, A5), and MATLAB analysis outcomes (A8). Images in (A6–A7) are non-to-scale schematics.

*Figure 2—figure supplement 4*: Single-plane laser confocal images shown as inverted grayscale images (A1–A2), thresholded images (A3, A4), and MATLAB analysis outcomes (A5, A6).

*Figure 3*: Maximum intensity projections of spinning disc confocal stacks through the ventral 1.2 µm of the cell for each time point (C–G).

*Figure 4*: Cells shown are maximum intensity projections of laser scanning confocal stacks through the ventral 1 µm of the cell. The tubulin channel (top) is shown as inverted grayscale images (A–C). Thresholded single focal plane images under the nucleus of the cell selected for directionality analysis (F–H).

*Figure 4—figure supplement 1*: (A–B) Cells shown are maximum intensity projections of laser scanning confocal stacks through the ventral 1 µm of the cell. The tubulin channel (top) is shown as inverted grayscale images. mCherry-labeled KIFDN constructs (bottom) are pseudo-colored magenta.

*Figure 5*: Cells shown are maximum intensity projections of laser scanning confocal stacks of the ventral 1 µm of the cell. The tubulin channel is shown as inverted grayscale images (A–B). Thresholded single focal plane images under the nucleus of the cell selected for directionality analysis (D–E).

*Figure 5—figure supplement 2*: Cells shown are maximum intensity projections of laser scanning confocal stacks of the bottom 1 µm of the cell. The tubulin channel is shown as inverted grayscale images.

Figure 6: Maximum intensity projections of spinning disc confocal stacks through the ventral 1.2 μm of the cell for each time point (A–B). Single focal plane spinning disc confocal time frames (D–E). For all images, whole-image contrast for each channel was adjusted equally within each figure.

## Quantitative image analysis
### Analysis of MT sliding after photobleaching
Maximum intensity projections of spinning disc confocal stack from the FRAP assay were analyzed. The photobleached cell areas outside the 'fluorescent belt' were thresholded to mask out cell background and insignificant amounts of low-signal MTs, which presumably represent polymerizing MT ends. The area of high-signal MTs moving into the bleached area from the fluorescent belt at every time point of the video was quantified as a proxy for MT sliding efficiency.

### SunTag rigor kinesin tracking analysis
The acquired image was processed through Imaris Microscopy Image Analysis Software (Oxford Instruments), where the fiducial marks were tracked. Utilizing the spot tracking feature, we filtered the fiducial marks based on intensity and employed the integrated Autoregressive Motion algorithm. Both the Max Distance and Gap sizes were enabled to precisely calibrate the tracks. Subsequently, we refined the tracks by filtering them based on intensity, thereby eliminating noise and particles that intermittently enter and exit focus. We sought to normalize the behaviors by comparing the MT movements over 5 s intervals and calculated the displacement of a given fiducial mark (see MATLAB scripts below). In total, over 60,000 tracks were detected, and the segmented displacement of ~25,000 of those tracks was calculated. The 5 s displacement was binned at 0.05 μm intervals, and the % of distribution for each bin was calculated for each cell and summarized in histograms (*Figure 1—figure supplement 1B*, *Figure 6F*). Displacements of less than 0.15 μm/5 s (below the resolution limit) were considered indicative of stationary MTs (*Figures 1J and 6G*). Displacements of greater than 0.3 μm/5 s were considered significant displacements indicative of motile MTs (*Figures 1K and 6H*).

### MATLAB script: MSDanalyzer, segmentation
The positions of all tracked fiducial spots were exported from Imaris to Excel. The MSDanalyzer (GitHub, @msdanalyzer) was used to normalize the tracks in time. Tracks were segmented into displacements over 5 s and binned as shown in the Results.

### MATLAB script: MT directionality
Oversampled images were deconvolved using the Richardson and Lucy Deconvolution algorithm. Images were masked and thresholded (IsoData) in ImageJ. The MT directionality script was applied in MATLAB (*Bracey et al., 2020*). Only the outer 1 μm of MTs were taken for binning and quantification purposes.

### KIF5B depletion quantification
Single focal plane laser scanning confocal images were quantified in ImageJ. Cell outlines were used as masks, and cells positive and negative for GFP expression (shRNA marker) were analyzed. Mean KIF5B immunostaining intensity per cell was measured. To avoid bias of potential staining fluctuation between data points, the intensity of each GFP-positive cell was normalized to the averaged intensity of GFP-negative cells in the same field of view.

## Statistics and reproducibility
For all experiments, n per group is as indicated by the figure legend and the scatter dot plots indicate the mean of each group and error bars indicate the standard error of the mean. All graphs and statistical analyses were generated using Excel (Microsoft) and Prism software (GraphPad). Statistical significance for all in vitro and in vivo assays was analyzed using an unpaired t-test, one-way ANOVA with Sidak's multiple comparisons test, Kolmogorov-Smirnov test as indicated in the figure legends.

For each analysis, p<0.05 was considered statistically significant, and *p<0.05, **p<0.01, ***p<0.001, ****p<0.0001.

## Materials availability statement

Materials developed in this study are available upon request from the corresponding author.

## Acknowledgements

This work was supported by National Institutes of Health (NIH) grants F31DK122650 T32 (to KMB), R35-GM127098 (to IK), R01-DK106228 (to IK and GG), R01-DK65949 (to GG), R01-DK125696 (to GG). KMB was supported by an NIH training grant R25-GM062459 'Initiative for Maximize Student Diversity' (Sealy, PI), AC by an NIH training grant T32-CA119925 'Integrated Biological Systems Training in Oncology' (Tansey, PI), CME was supported by NIGMS of the NIH under award number T32GM007347, and PN by an NIH training grant T32 DK101003 'Integrated Training in Engineering and Diabetes' (Young, PI). We thank Hamida Ahmed for technical help.

## Additional information

### Funding

| Funder | Grant reference number | Author |
|---|---|---|
| National Institutes of Health | F31DK122650 | Kai M Bracey |
| National Institutes of Health | R35GM127098 | Irina Kaverina |
| National Institutes of Health | R01DK106228 | Guoqiang Gu Irina Kaverina |
| National Institutes of Health | R01DK65949 | Guoqiang Gu |
| National Institutes of Health | R01DK125696 | Guoqiang Gu |

The funders had no role in study design, data collection and interpretation, or the decision to submit the work for publication.

### Author contributions

Kai M Bracey, Formal analysis, Investigation, Methodology, Writing – original draft; Margret A Fye, Kung-Hsien Ho, Formal analysis, Investigation, Methodology; Alisa Cario, Pi'illani Noguchi, Formal analysis; Guoqiang Gu, Conceptualization, Supervision, Funding acquisition, Validation, Writing – review and editing; Irina Kaverina, Conceptualization, Resources, Supervision, Funding acquisition, Validation, Methodology, Writing – review and editing

### Author ORCIDs

Kai M Bracey (ORCID) https://orcid.org/0009-0001-5483-0718
Irina Kaverina (ORCID) https://orcid.org/0000-0002-4002-8599

### Ethics

This study was performed in strict accordance with the recommendations in the Guide for the Care and Use of Laboratory Animals of the National Institutes of Health. All of the animals were handled according to approved institutional animal care and use committee (IACUC) protocol M1800195-02 of Vanderbilt University.

Joint Public Review: https://doi.org/10.7554/eLife.89596.4.sa1
Author response https://doi.org/10.7554/eLife.89596.4.sa2

## Additional files

### Supplementary files
MDAR checklist

### Data availability
The numeric data used for this study are available as source data spreadsheets.

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
