## [Editor Report · eLife Assessment]

In their **valuable** study, Bracey et al. investigate how microtubule organization within pancreatic islet beta cells supports optimal insulin secretion. Using a combination of live imaging and photo-kinetic assays in an in vitro culture system, they provide **compelling** evidence that kinesin-1-mediated microtubule sliding, which plays critical roles in neurons and embryos, also plays a critical role in forming the sub-membranous microtubule band in response to glucose in beta cells. This work will be of interest to cell biologists studying cytoskeletal dynamics and organelle trafficking, as well as to translational biologists focused on diabetes.

---

## [Referee Report · Joint Public Review]

This elegant study provides important insights into the organization of sub-membrane microtubules in pancreatic β-cells, highlighting a key role for the motor protein KIF5B. The authors propose that KIF5B drives microtubule sliding and alignment along the plasma membrane, a process enhanced by high glucose levels. This precise microtubule arrangement is essential for regulated secretion in β-cells. Supporting this model, the authors show that KIF5B is more highly expressed than other kinesins in MIN6 cells, and its depletion via shRNA disrupts sub-membrane microtubule density and organization. In contrast, KIF5A knockdown alters overall microtubule architecture. Using a dominant-negative approach, they further demonstrate that KIF5B-mediated microtubule sliding relies on its tail domain and is stimulated by glucose, paralleling known glucose-dependent increases in kinesin-1 activity.

---

## [Author Response]

The following is the authors’ response to the previous reviews

**Reviewer #1 (Public review):**
Specific comments:(1) It is difficult to appreciate that there is a "peripheral sub-membrane microtubule array" as it is not well defined in the manuscript. This reviewer assumes that this is in the respective field clear. Yet, while it is appreciated that there is an increased amount of MTs close to the cytoplasmic membrane, the densities appear very variable along the membrane. Please provide a clear description in the Introduction what is meant with "peripheral sub-membrane microtubule array".

A definition has been added to the Introduction.

(2) The authors described a "consistent presence of a significant peripheral array in the C57BL/ 6J control mice, while the KO counterparts exhibited a partial loss of this peripheral bundle.Specifically, the measured tubulin intensity at the cell periphery was significantly reduced in the KO mice compared to their wild-type counterparts". In vitro "control cells had convoluted nonradial MTs with a prominent sub-membrane array, typical for β cells (Fig. 2A), KIF5B-depleted cells featured extra-dense MTs in the cell center and sparse receding MTs at the periphery (Fig. 2B,C)". Please comment/discuss why in vivo there are no "extra-dense MTs in the cell center".

We now add a discussion of this point, which we believe could be a manifestation of 3D shape of a beta cell in tissue and/or compensatory mechanisms in organisms.

(3) Authors should include in the Discussion a paragraph discussing the fact that small changes in MT configuration can have strong effects.

A paragraph added to the discussion.

**Recommendations for the authors:**

**Reviewer #1 (Recommendations for the authors):**
(1) Figure 1: Even though the reviewer appreciates that minor changes of MT configuration have severe effects, still the overall effects appear minor (40 vs. <50% or 35% vs. around 28%). Notably, there are no statistically significant differences in the different groups in Fig. 1Suppl-Fig.1 D. This reviewer is not sure if the combination of many not significantly different data points can result in significant changes and this should be checked by a statistician. Authors should include in the Discussion a paragraph discussing the fact that small changes in MT configuration can have strong effects.

We have now added the requested paragraph to the discussion. Indeed, the differences are small, and the significance is only detected in a data set with a large sample size in Fig. 1J,K (combined data sets with smaller sizes from Fig. 1-Suppl-Fig.1 D), consistent with the fact that a larger sample size generally provides more power to detect an effect.

(2) Unfortunately, the authors cannot block kinesin-1 resulting in microtubule accumulation in the cell center and then release the block (best inhibiting microtubule formation), to show that the MTs accumulated in the cell center will be transported to the periphery.

This is indeed the case at the moment, yes.

Minor comments:- Abstract: β-cells vs. β cells (and throughout the manuscript)- Page 4: "MTOC, the Golgi, (Trogden et al. 2019), and"- Page 5: "β-cell specific"- MT-sliding vs. MT sliding- Kinesin 1 vs. kinesin-1- Page 6, line 1: "β cells. actively"- Page 7: "a microtubule probe", should be "MT"- Page 9: "1μm" vs. "1 μm"- Page 10: "demonstrate a dramatic effect" recommended is: "demonstrate a marked effect"- Page 13, line 1: dramatically vs. markedly- Page 13, line 5: "50μm" vs. "50 μm" (in general, there should be a space between number and unit?)- "37 degrees C" vs. "37{degree sign}C"- Animal protocol number?- "Mice were euthanized by isoflurane inhalation"? What concentration? How long? More details are needed (no cervical dislocation?).- Antibodies: more identifiers are needed.- Antibody information in Key reagents and under 5. Reagents and antibodies do not fit (1:500 and 1:1000).

Thank you, we corrected all relevant information now.